# Estimating Dietary Protein and Sodium Intake with Sodium Removal in Peritoneal Dialysis Patients

**DOI:** 10.3390/metabo14080460

**Published:** 2024-08-19

**Authors:** Ana Bontić, Aleksandra Kezić, Jelena Pavlović, Marko Baralić, Selena Gajić, Kristina Petrovic, Vidna Karadžić Ristanović, Olga Petrović, Vera Stjepanović, Sanja Stanković, Milan Radović

**Affiliations:** 1Clinic for Nephrology, University Clinical Center of Serbia, Pasterova 2, 11000 Belgrade, Serbia; aleksandra.kezic@med.bg.ac.rs (A.K.); jelena@pavlovic.rs (J.P.); marko.baralic@med.bg.ac.rs (M.B.); selenagajic@gmail.com (S.G.); kikapetrovic123@gmail.com (K.P.); vidnakaradzic@gmail.com (V.K.R.); milan.radovic@med.bg.ac.rs (M.R.); 2Faculty of Medicine, University of Belgrade, Doktora Subotica Starijeg 8, 11000 Belgrade, Serbia; olga.petrovic@med.bg.ac.rs; 3Clinic for Cardiology, University Clinical Center of Serbia, Pasterova 2, 11000 Belgrade, Serbia; 4Center for Medical Biochemistry, University Clinical Center of Serbia, Pasterova 2, 11000 Belgrade, Serbia; verastj@gmail.com (V.S.); sanjast2013@gmail.com (S.S.); 5Faculty of Medical Sciences, University of Kragujevac, Svetozara Markovica 69, 34000 Kragujevac, Serbia

**Keywords:** peritoneal dialysis, dietary protein intake, sodium intake, sodium removal, comorbidity, residual renal function

## Abstract

An increase in dietary protein intake (DPI) carries a risk with respect to increased sodium intake, which further leads to the development of cardiovascular morbidity in peritoneal dialysis (PD) patients. Dialytic (DSR) and urinary sodium removal (USR) are potential indicators of sodium intake. In this single-center cross-sectional study with 60 prevalent PD patients, we analyze the correlation of DPI with sodium intake and the association between residual renal function (RRF) and comorbidity grade, expressed as the Davies score with sodium removal and protein metabolism indices such as normalized protein catabolic rate (nPCR) and lean body mass (LBM). The value of RRF < 2 mL/min/1.73 m^2^ is significantly associated with lower USR (*p* = 0.000) and lower %LBM (*p* < 0.001). The greatest USR is detected in patients with low Davies comorbidity grade (*p* = 0.018). Compared to patients with DPI < 0.8 g/kg/day, patients with DPI > 0.8 g/kg/day have a greater sodium intake (3.69 ± 0.71 vs. 2.94 ± 0.86; *p* < 0.018) and a greater nPCR (*p* < 0.001). Protein intake is significantly correlated with sodium intake (*p* = 0.041), but not with total sodium removal (TSR). A strong correlation is observed between sodium intake and TSR (*p* = 0.000), although single TSR values are not the same as the corresponding sodium intake values. An increasing protein intake implies the necessity to determine both sodium intake and sodium removal. Preservation of RRF has a beneficial role not just in sodium removal, but also in the increase of LBM.

## 1. Introduction

Cardiovascular diseases are the most frequent causes of comorbidity and mortality in peritoneal dialysis (PD) patients. The majority of these patients are volume-overloaded. Therefore, maintenance of an optimal volemia is one of the most important goals of managing PD patients [1,2]. Achieving euvolemia is the final result in the equation comprising salt and water intake together with residual diuresis (RD) and optimal ultrafiltration (UF). All of these parameters are necessary for blood pressure control, reduction of left ventricular hypertrophy (LVH), prevention of cerebrovascular risks, and patient survival [3,4,5,6]. The European automated peritoneal dialysis outcome study (EAPOS) has shown that the failure to achieve > 750 mL of daily UF in anuric patients is associated with increased mortality [5].

Although the results concerning the prediction of sodium removal on the survival of PD patients are not consistent, the importance of determining sodium removal is recognized [7,8]. In addition to extracellular water (ECW) expansion, sodium itself exerts detrimental effects by accumulating in the interstitial tissue and leading to salt-sensitive hypertension and accelerated loss of residual renal function (RRF) [9]. This effect is important, since the rate of RRF decline is a powerful prognostic factor for PD patient survival and impacts negatively urinary sodium removal (USR) [10].

Because of the aforementioned reasons, it is recommended that the daily intake of sodium in patients on peritoneal dialysis should not exceed 2.3 g [11]. To some extent, an increased intake of protein-rich foods is associated with a higher sodium intake [12,13]. This can further complicate the management of dialysis patients, considering that the reduction of protein intake with inadequate dialysis affects the development of sarcopenia and protein–energy wasting (PEW) [14,15]. This aggravated protein catabolism starts with the decline of the renal function via a complex mechanism. Reduced lean body mass (LBM) is one of the indicators of reduced somatic protein storage and PEW, and is predictive of mortality [16]. Management of patients with CKD stage IV implies a significant reduction of protein intake, but the recommended daily protein dose for peritoneal dialysis patients increases to 1.2 g/kg of their body weight [17]. However, this target is practically hardly ever achieved, and a substantial number of studies reports that even a lower dietary protein intake (DPI) could maintain good nutrition, but that a level of DPI < 0.8 g/kg/day carries a great risk for PEW and worst outcome [17,18]. In addition to DPI estimated by food diaries or dietary recall, a protein equivalent of nitrogen appearance (PNA), also known as a normalized protein catabolic rate (nPCR), is highly recommended in the assessment of nutrition and protein metabolism in PD patients [19]. Calculation of nPCR is based on urinary and peritoneal urea losses. Similarly, the sodium loss calculated using the urine and peritoneal dialysis effluent is a good surrogate for sodium intake estimated by food diaries or dietary recall.

From all the above, it is necessary to confirm that the importance of RRF is not just related to the removal of water and sodium, but also to the visceral protein store in PD patients. Patient adherence to a diet and a controlled sodium intake is a challenging task when managing PD patients. A special review should be made about the addition of salt to food, a phenomenon which can also be understood as being a feature of the national cuisine.

Bearing in mind the complex mechanisms that influence the occurrence of protein malnutrition in CKD patients with a tendency toward sodium retention, this study aims to explore a cohort of PD patients with respect to the association of sodium handling and protein metabolism indices, with RRF and the comorbidity grade expressed as the Davies score. We also explore whether salt consumption is correlated with protein intake.

## 2. Materials and Methods

This is a single-center cross-sectional study with 60 prevalent PD patients over the age of 18 years, treated in the Clinic for Nephrology at the University Clinical Center of Serbia. All procedures were in accordance with the Helsinki Declaration. The study protocol was approved by the ethics committee of the Faculty of Medicine, University of Belgrade (61206-3090/2/16). All participants signed an informed consent to participate in the study. The main investigated variables, DSR, USR, nPCR, LBM, dietary sodium intake, DPI, and dietary energy intake (DEI), were analyzed with other clinical and laboratory variables (gender, age, Hb, C-reactive protein, creatinine, sodium, albumin) and with dialysis adequacy parameters (Kt/V, weekly creatinine clearance-CCl), peritoneal transport characteristics (D/PCr), PD modalities, RD, body mass index (BMI), mean arterial pressure (MAP), and Davies comorbidity index (DCI).

### 2.1. Laboratory Method

Samples of daily dialysate were collected from patients to determine sodium removal. DSR was calculated as the difference between the concentration of sodium in the drained dialysate multiplied by the total volume output and the concentration of sodium in the instilled dialysate multiplied by the total filling volume. USR was calculated by multiplying the sodium concentration in daily urine, expressed as mmol per liter, by the daily urine volume expressed in liters. Total sodium removal (TSR) was calculated as the sum of DSR and USR. Sodium removal was measured using the indirect ion selective electrode method (I-ISE) on an Abbott Architect ci8200 analyzer (Abbott Diagnostics, Wiesbaden, Germany) and the other biochemical in daily sample of dialysate and urine before the PET test.

A standard PET was performed. Dialysate samples were collected at 0 min, 120 min, and 240 min, and a blood sample was collected. The D/P creatinine was calculated as the ratio of the dialysate concentration of creatinine at 240 min to the serum concentration, the D/D0 glucose was calculated as the ratio of dialysate concentration of glucose at 240 min to that at time 0 [20,21]. The patients were assigned to one of four groups: S—slow, SA—slow average, F—fast, FA—fast average. RRF was calculated as the mean of creatinine and urea clearance corrected for the body surface area (mL/min per 1.73 m^2^).

Serum albumins were measured using the bromocresol purple (BCP) colorimetric method assay (Architect c 8000, Abbott Laboratories, Chicago, IL, USA). Albumins in the peritoneal effluent were measured by using the bromocresol green colorimetric assay (BCG), which forms a green complex with albumin (Beckman Coulter AU analyzer, Brea, CA, USA). Proteins in the urine and peritoneal effluent were determined using the colorimetric method using pyrogallol red combined with molybdate (Beckman Coulter AU analyzer).

Dialysis adequacy was expressed by urea clearance (Kt/V) and weekly creatinine clearance (weekly CCI). The Baxter software package PD Adequetest 2.0 (Healthcare, Deerfield, IL, USA) was used for the assessment of Kt/V, CrCl, and nPCR.

Comorbidity was assessed using the Davies comorbidity index [22]. This index was scored based on the presence of seven comorbid conditions: ischemic heart disease, peripheral vascular disease, left ventricular dysfunction, diabetes mellitus type 1 or type 2, systemic collagen vascular disease, malignancy, and any other condition that may affect overall survival such as cirrhosis, chronic obstructive pulmonary disease-COPD, and psychosis. The presence of each disease was scored with 1, and the absence with 0. The value of the score ranged from 0–7, and patients were divided according to risk groups: low risk (score value 0), medium risk (score 1 and 2), high risk (score 3 to 7).

The total dietary intake of sodium, protein, and energy was assessed through a 3-day dietary recall. Then, DPI, expressed as gram of proteins/kg of body weight, and dietary energy intake (DEI), expressed as kcal/kg of body wight, were calculated as average daily intake. Calculation of the total calorie intake was based on the sum of the intake from dietary and peritoneal dialysate.

Sodium intake was calculated as the average daily ingested amount expressed in mmol. Additionally, LBM was measured using creatinine kinetics. LBM was calculated according to the following formula [16]:LBM = 7.38 + 3.29 (CE + CD)

CE is the abbreviation for creatinine excretion in millimoles per day and CD is the abbreviation for creatinine degradation in millimoles per day. CE was calculated as the sum of the urinary creatinine output in millimoles per day (UCO) and the dialysate creatinine output in millimoles per day (DCO).
CE = UCO + DCO

CD was calculated using the following formula:CD = 0.04 × [Cr]plasma × body weight

Plasma creatinine concentration ([Cr]plasma) was expressed in µmol/L. LBM was normalized to the ideal body weight (IBW) and subsequently represented as %LBM. IBW was calculated as height (cm) − 105.

### 2.2. Statistical Methods

Continuous variables were expressed as the mean (standard deviation), if they were normally distributed, or as the median (interquartile range), if they exhibited other types of distribution. Frequencies were given as a percentage and number. The Shapiro–Wilk test was used to test for the normality of the data. Differences in continuous variables among the analyzed groups were tested using parametric tests (Student’s *t*-test and ANOVA with post-hoc analysis performed with the Bonferroni correction) and non-parametric tests (Mann–Whitney U test, Kruskal–Wallis test). For categorical variables, the Chi-square test was used. The Pearson’s correlation analysis was used to analyze the relationship of the LBM, DPI, protein intake, and TSR with other investigated parameters. The results are presented in tables and through graphic illustrations. All data were processed in the SPSS 20.0 (IBM Corp. Released 2011. IBM SPSS Statistics for Windows, Version 20.0. Armonk, NY, USA: IBM Corp.) software package and R 3.4.2 (R Core Team (2017). R: A language and environment for statistical computing. R Foundation for Statistical Computing, Vienna, Austria). Values of *p* < 0.05 were accepted as significant.

## 3. Results

Demographic data, laboratory values, and clinical characteristics of all the patients, divided in two groups according to RRF lower or greater than 2 mL/min/1.73 m^2^, are presented in Table 1.

The patients were treated using different PD regimes (CAPD, CCPD, APD) for 3–147 months (data not presented in a table). There were nine (32.1%) patients with diabetes mellitus (DM) in group 1 (RRF < 2 mL/min/1.73 m^2^), and seven (21.9%) in group 2 (RRF > 2 mL/min/1.73 m^2^). Between them, there were no differences regarding age, sex, DM, Davies comorbidity grade, transport status of peritoneal membrane prevalence, small solute peritoneal transport characteristics (D/Pcr), and BMI.

There was no difference in the distribution of all the PD regimes between group 1 and group 2. Patients in group 1 had significantly lower residual diuresis (*p* = 0.00) and, consequently, USR (*p* = 0.00). Concomitantly, in group 1, there was a significantly higher incidence of DSR (*p* < 0.021) and UF (*p* < 0.004).

Overall, patients in group 2 had significantly greater TSR (*p* < 0.046). Total mass removal of sodium in group 1 was 4.58 g/day and, in group 2, it was 5.83 g/day. There were no differences in dietary energy intake and dietary intake of sodium and protein, but patients in group 1 had significantly lower %LBM (*p* < 0.001).

Differences in the investigated parameters in relation to the comorbidity grade, classified as low, medium, and high, are presented in Table 2.

The differences in age, PD duration, MAP, eGFR, protein intake, D/PCr, UF, DSR, and TSR were not significant. There was a highly significant difference between the groups according to the Davies comorbidity grade in RD, showing the greatest value of RD in patients with a low comorbidity grade. Also, the greatest value of USR was detected in the low comorbidity grade group of patients. Patients with a medium comorbidity grade exhibited significantly greater values of sodium intake (4.02 ± 0.58 g/day; *p* = 0.008) compared to the low comorbidity grade group (3.12 ± 0.86 g/day). Regarding dietary energy and protein intake, there were no significant differences between groups. Additionally, a significant difference was noticed in dialysis adequacy parameters Kt/V and weekly CrCl.

Patients with the lowest comorbidity grade had the greatest serum albumin values and Hb value, but the lowest CRP value, as depicted in Table 2. A significant difference in the prevalence of LVH was observed, with the highest prevalence found in the group of patients exhibiting the highest comorbidity grade.

The influence of DPI lower or greater than 0.8 g/kg/day and its association with the investigated parameters are shown in Table 3.

Patients with a DPI > 0.8 g/kg/day exhibited a greater caloric intake (*p* < 0.035) and sodium intake (*p* < 0.018). There were no significant differences in age, serum albumins, %LBM, blood cholesterol level, PD adequacy indices (Kt/V, CrCl), and peritoneal protein loss.

The relationships among DPI, protein intake, and other nutrition parameters were investigated in male and female subjects. Following a correlation analysis between DPI and other nutrition parameters, a strong association was noticed with DEI in both male (*p* = 0.002) and female (*p* = 0.00) patients (Table 4). Also, protein intake correlated positively with total energy intake in both sexes (*p* = 0.031 for males and *p* = 0.006 for females). In female patients, a significant positive correlation between DPI and nPCR (*p* = 0.009) was found; in males, this correlation was borderline significant (*p* = 0.067). DPI negatively correlated with BMI in both males and females (*p* = 0.021 and *p* = 0.04, respectively). No correlation was observed with %LBM and albumins.

In addition to a significant positive correlation between %LBM and albumins (*p* = 0.003), a strong association was also noticed with RRF (*p* = 0.003) and CCL (*p* = 0.007) (Table 5). Additionally, the female gender tended to exhibit lower values of LBM (*p* = 0.015). No association was found with age, Kt/V, nPCR, and peritoneal protein loss.

A highly significant correlation was found between sodium intake and total sodium removal (Figure 1).

A significant correlation was found between sodium intake and protein intake (Figure 2).

Although protein intake correlates with sodium intake, no association was found between protein metabolism indices and TSR: DPI (*p* = 0.408), TPI (*p* = 0.147), and nPCR (*p* = 0.336) (Table 6).

## 4. Discussion

Our results confirm the importance of RRF not just in the removal of water and sodium, but also with respect to its association (when preserved) with protein storage in PD patients. These results are not surprising and are in line with results from previous studies. Preserved RRF is associated with a lower degree of inflammation, better removal of uremic toxins of a higher molecular weight, better nutritional status, and better survival in both PD and HD patients [23,24,25]. In light of already known facts, the findings from our study are clear, showing higher values of Hb and albumin with lower CRP values in patients with the lowest Davies comorbidity score compared to patients with the highest comorbidity score.

Patients with a lower diuresis and a lower urinary sodium daily loss were associated with a greater Davies comorbidity grade. Additionally, the group of patients with a high comorbidity grade exhibited the greatest prevalence of LVH. The association between declining RRF and LVH and increased cardiovascular morbidity and mortality is well known [26,27,28,29]. The more severe the diuresis, the better the volume control with a lower comorbidity grade. A volume-overloaded patient, usually, is a patient with an increased body sodium content. However, the question arises as to whether sodium has a direct toxic effect, in addition to the indirect effect, by inducing hypervolemia. Might sodium have an influence in the development of comorbid conditions independently of extracellular volume regulation? Using Na^23^-MRI, it has been shown that sodium stored in the interstitial skin tissue not only correlates with BP levels, but progressively increases as the eGFR declines [30,31]. Furthermore, skin sodium has been shown to correlate with the left ventricular mass irrespective of fluid overload [29]. It has also been shown that a higher sodium content in the skin is accompanied by increased inflammatory parameters in the serum, such as CRP and IL-6 [32]. Regardless of the RRF degree, a significantly higher intake of sodium was observed in our patients and, consequently, increased removal of sodium by urine and peritoneal dialysis. Although sodium intake correlated with the total protein intake, which is generally the case with the diet in Western societies, the salt intake in our patients was even greater, a phenomenon which clearly indicates a higher intake of processed foods with a high sodium content. The sodium intake was assessed by recall method, which is prone to errors, but TSR was found to be highly correlated with sodium intake and was notably higher than the values published in previous studies [33,34]. TSR, as the sum of USR and DSR, can be regarded as a relatively precise estimate of salt intake. It is influenced by dialysis prescription and diuretic use. By comparing TSR with the corresponding sodium intake values, we can conclude that TSR values were inadequately higher. Similar results have been observed in other studies and it has been suggested that this difference between sodium intake and sodium removal might be partly the consequence of an underreporting of portion size or even of the composition of the food that was ingested [13,33,35]. To understand the reason behind the obvious gap between sodium intake and sodium removal in the dialysis population, a well-designed sodium balance study needs to be conducted.

Nonetheless, the dietary recall method showed a positive association between sodium and total protein intake, although not a strong one. Additionally, protein intake was not statistically different among patients with different comorbidity grades, but a difference was observed regarding sodium intake. It implies salt addition to the food. Unlike sodium intake, protein intake was slightly lower than the recommended amounts in PD patients. Dietary protein intake correlated with dietary energy intake and nPNA, but not with albumins and LBM. Indeed, in CKD patients, the dietary protein intake can be efficiently estimated by measuring the urea nitrogen appearance, requiring 24 h of urine collection [36]. nPNA, or nPCR, is an equivalent of DPI and, as a measure of nutritional status in PD patients, is assessed using a calculation based on urea appearance in both 24 h urine and 24 h dialysate effluent. nPCR may be even superior for predicting CAPD dialysis adequacy, pointing out the close association between nutritional status and dialysis adequacy [37]. Our female subjects showed a strong correlation between DPI and nPCR, such a correlation failing to be observed in men, although it has tendency to be significant. This may be a consequence of the relatively small number of subjects included in our study or of imprecise dietary estimates.

Unintentional low DPI of 0.8 g/kg/day for at least 2 months in dialysis patients is a criterion for PEW. Accordingly, the cut-off value for DPI in our study for analysis was 0.8 g/kg/day. Dialysis patients can maintain a good nutrition status with 0.8 g/kg/day of DPI, but some studies have shown that only DPI values of 0.73 g/kg/day and below are associated with reduced survival in PD patients [18,38]. A diagnosis of malnutrition is critical in dialysis patients, since malnutrition is associated with high morbidity, mainly cardiovascular, and mortality [39,40]. PEW is the state of decreased body protein and energy fuel storage (muscle and fat masses). Unlike malnutrition, in which abnormalities are induced by an inadequate diet, PEW refers to abnormalities that cannot be corrected solely by changing the diet and are induced, usually, by inflammation and accompanied by decreased serum albumin levels [14]. LBM, as an indicator of somatic protein storage, is commonly used as a nutritional index in PD patients, but there is some disagreement regarding its use. Partly, the cause of this problem is the type of method used to determine LBM, i.e., whether the tracer (antipyrine) dilution method is used, or anthropometry, bioelectrical impedance, or the creatinine kinetics method (LBM-CK) [41,42].

In our study, we determined LBM by using creatinine kinetics. Our results reveal, except for the already mentioned lack of correlation with DPI, a strong association of LBM with RRF, BMI, and serum albumins. Patients belonging to the group with RRF > 2 mL/min/1.73 m^2^ had a significantly greater value of LBM. This finding was expected and is consistent with the results from other large studies that have shown a positive relationship between preserved RRF and an improved nutritional status [43,44]. The decline in RRF was accompanied by a decline in LBM values [25]. The absence of differences in other parameters of nutritional status such as BMI, nPCR, cholesterol, and albumin, based on a value of RRF greater or lower than 2 mL/min/1.73 m^2^, may be a consequence of the small number of subjects, but it is also an indicator of an adequate dialysis treatment for our patients. Despite the differences in total Kt/V and CrCl, target values were achieved regardless of the RRF value, and this had a favorable effect on patients’ appetite and nutrient intake.

A positive correlation between LBM and serum albumin has also been found in other studies, but there are also studies reporting no association of LBM with serum albumin [41,45]. A possible explanation for these conflicting findings could be the impact of inflammation, which affects, to different extents, serum albumin levels and somatic protein storage. Since LBM is a non-fat component, it is clear why a negative correlation was observed with the female gender, the latter having a higher fat estimate than the male gender. Our results also show that patients who had been subject to the PD treatment for longer ate more and exhibited lower BMI values than patients who had been treated with PD for less than a year. One explanation might be that PD patients continue to eat less protein-rich food according to the recommendations received during the pre-dialysis period. However, these results should be confirmed by investigating a larger group of subjects who are equally represented based on DPI values lower or greater than 0.8 g/kg/day.

Regardless of the presence of diagnosed comorbid conditions, or regardless of a lower RRF degree and protein intake than that that is recommended, our subjects consumed an impermissibly high amount of salt. One might consider them as non-adherent to the dietary prescription. Although the total protein intake correlated with sodium intake, this association was not particularly strong, and a correlation between DPI and sodium intake was not observed. It is obvious that there is a problem with respect to the addition of salt to food, a habit which can also be understood as being a feature of the national cuisine and eating habits. Dietary habits differ among ethnic groups, a phenomenon which should be taken into account in everyday practice [46,47]. Instead of increasing their protein intake and reducing their extra salt intake, our subjects primarily ate extremely salty foods. It is necessary for patients to undergo training regarding dietary habits as often as possible and to be encouraged to keep food diaries in addition to preparing home meals without the use of extra salt and processed foods. Additionally, longer conversations during visits in hospital conditions and the activation of telemedicine visits are also recommended.

There are several limitations in our study, such as the relatively small sample size, as well as the single-center study design, the lack of repeated measurements, and the conduction of our analysis in a prospective manner based on potential confounding factors such as age, gender, and baseline nutritional status. Nevertheless, our findings emphasize the significance of RRF preservation not just for sodium removal, but also to increase the LBM. An increase in the value of nPCR, which is a reliable indicator of protein intake, indicates the need to determine the intake and removal of sodium. It remains to be determined in survival studies adopting a prospective design and including a larger number of patients what TSR values should be regarded as adequate. Definitely, in patients without RRF, it is necessary to implement a PD regimen that maximally removes sodium.

## 5. Conclusions

Protein intake values determined either as a result of an increase in nPCR or by checking dietary habits indicate the necessity to determine both sodium intake and sodium removal. A practical approach might be the use of the best available tool (food intake diary, food sodium content stickers, or similar) in order to discriminate between pure high protein intake and additional salt load. LBM is just one of many nutritional indicators positively associated with preserved RRF, whose monitoring in patients with severe cardiovascular comorbidities involves the measuring of the urinary content of sodium. The involvement of dietitians in the daily work of nephrologists, in addition to checking patient adherence to dietary recommendations, is of crucial importance when managing PD patients.

## Figures and Tables

**Figure 1 metabolites-14-00460-f001:**
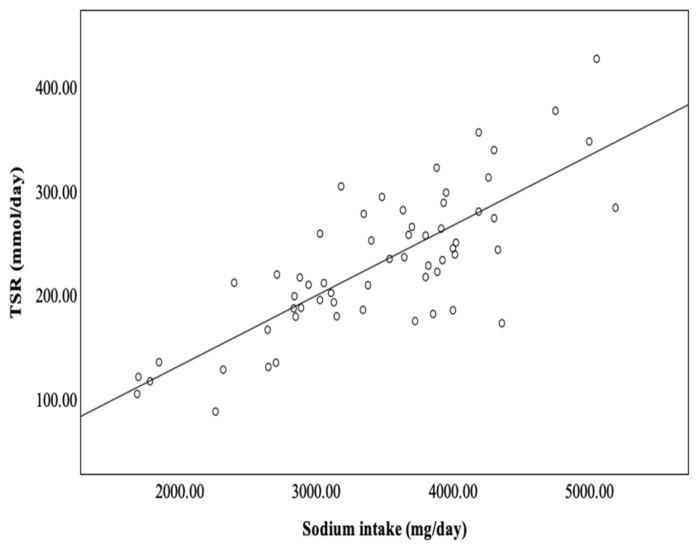
Correlation between daily sodium intake and daily total sodium removal (TSR) in PD patients (r = 0.784; *p* = 0.000).

**Figure 2 metabolites-14-00460-f002:**
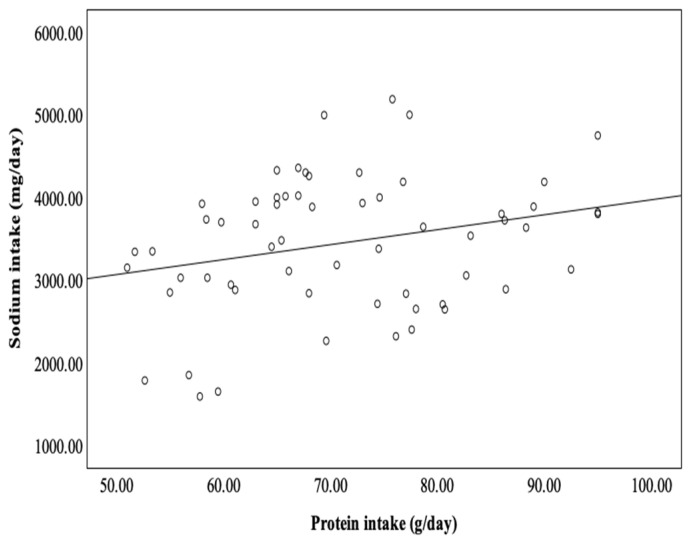
Correlation between daily protein and sodium intake in PD patients (r = 0.267; *p* = 0.041).

**Table 1 metabolites-14-00460-t001:** Patient characteristics according to the values of RRF lower or greater than 2 mL/min/1.73 m^2^.

	RRF < 2 mL/min/1.73 m^2^ (n = 28)	RRF > 2 mL/min/1.73 m^2^ (n = 32)	*p*-Value
Age (years)	56.39 ± 13.25	58.28 ± 12.53	0.573
Male gender (n; %)	11 (39.3%)	17 (53.1%)	0.208
BMI (kg/m^2^)	24.87 ± 3.57	24.92 ± 4.73	0.930
%LBM	37.23 ± 7.14	51.48 ± 11.41	**<0.001**
PD duration (months)	51 (3–147)	24.75 (3–67)	**0.001**
CAPD vs. CCPD/APD (n; %)	20 (71.4%)	22 (68.8%)	0.523
DM (n; %)	9 (32.1%)	7 (21.9%)	0.272
Davies comorbidity grade low/(medium + high) (n; %)	8 (28.6%)	12 (37.5%)	0.325
D/PCr	0.67 ± 0.11	0.64 ± 0.12	0.259
F/FA transporters (n; %)	19 (67.9%)	16 (50%)	0.128
RD (mL/day)	800 (0–1650)	1425 (300–3235)	**0.000**
UF (mL/day)	1197.32 ± 440.35	866.09 ± 425.65	**<0.004**
MAP (mmHg)	98.28 ± 10.64	92.9 ± 9.65	**<0.046**
Kt/V	2.2 (1.3–3.33)	2.61 (1.6–4.4)	**0.011**
CrCl (L/week)	59.85 (44–115.9)	95.45 (55.8–233.6)	**0.000**
Hb (g/dL)	8.5 (9–11.8)	9.75 (8.9–14)	**0.03**
Creatinine (mg/dL)	9.2 ± 2.17	6.67 ± 1.7	**<0.001**
Albumin (g/L)	32.61 ± 5.98	35.03 ± 5.23	0.099
Cholesterol (mg/dL)	5.46 ± 1.74	5.01 ± 1.3	0.465
CRP (mg/L)	5.95 (0.6–23)	4.98 (0.3–14.6)	0.101
Sodium intake (g/day)	3.51 ± 0.84	3.46 ± 0.83	0.883
Protein intake (g/day)	66.71 ± 14.13	71.65 ± 13.03	0.373
DPI (g/kg/day)	0.96 ± 0.15	0.93 ± 0.2	0.65
nPCR (g/kg/day)	0.86 ± 0.23	0.84 ± 0.21	0.836
DEI (kcal/kg/day)	29.63 ± 2.7	26.81 ± 6.52	0.249
DSR (mmol/day)	171.7 ± 57.97	132.77 ± 64.7	**<0.021**
USR (mmol/day)	72.8 (0–165)	101.15 (25.2–338.1)	**0.000**
TSR (mmol/day)	211.07 ± 62.81	253.54 ± 92.56	**<0.046**

MAP = mean arterial pressure; DM = diabetes mellitus; RD = residual diuresis; UF = ultrafiltration; Hb = hemoglobin; CRP = C-reactive protein; CAPD = continuous ambulatory peritoneal dialysis; CCPD = continuous cyclic peritoneal dialysis; APD = automated peritoneal dialysis; Kt/V = dialysis adequacy; CrCl = creatinine clearance; BMI = body mass index; LBM = lean body mass; D/Pcr = dialysate/plasma creatinine; F/FA = fast/fast-average; DPI = dietary protein intake; DEI = dietary energy intake; nPCR = normalized protein catabolic rate; DSR = dialytic sodium removal; USR = urinary sodium removal; TSR = total sodium removal.

**Table 2 metabolites-14-00460-t002:** Differences in clinical parameters depending on Davies comorbidity grade (DCG).

	DCG—Low	DCG—Medium	DCG—High
	(n = 19)	(n = 28)	(n = 13)
Age (years; mean ± SD)	56.63 ± 11.54	55.50 ± 13.25	64.08 ± 12.18
PD duration (months; mean ± SD)	33.42 ± 33.81	32.71 ± 41.53	27.61 ± 20
Sodium intake (g/day; mean ± SD)	3.12 ± 0.86 ^++^	4.02 ± 0.58	3.15 ± 0.25
Protein intake (g/day; mean ± SD)	68.85 ± 12.25	75.97 ± 13.34	59.31 ± 10.30
DPI (g/kg/day; mean ± SD)	0.91 ± 0.18	1 ± 0.21	0.82 ± 0.12
nPCR (g/kg/day; mean ± SD)	0.87 ± 0.23	0.83 ± 0.21	0.79 ± 0.16
DEI (kcal/kg/day; mean ± SD)	26.16 ± 5.01	30.32 ± 6.49	24.34 ± 4.46
DSR (mmol/day; mean ± SD)	149.45 ± 61.50	152.27 ± 64.68	151.73 ± 71.72
USR (mmol/day; mean ± SD)	113.64 ± 83.95 **	78.93 ± 67.37	43.55 ± 33.86
TSR (mmol/day; mean ± SD)	263.12 ± 94.25	230.68 ± 77.77	195.29 ± 54.6
D/PCr (mean ± SD)	0.61 ± 0.12	0.67 ± 0.12	0.70 ± 0.09
Kt/V (mean ± SD)	2.63 ± 0.73 **	2.53 ± 0.62 **	1.88 ± 0.40
CrCl			
(L/week; mean ± SD)	90.99 ± 44.3 *	89.56 ± 28.91 *	60.32 ± 12.84
UF			
(mL/day; mean ± SD)	1077.63 ± 443.45	970 ± 480.66	1046.54 ± 62.84
RD			
(mL/day; mean ± SD)	1339.74 ± 814.47 **	942.86 ± 662.87	560.77 ± 428.57
Cr (mg/dL; mean ± SD)	8.23 ± 2.46	7.35 ± 2.34	8.37 ± 1.88
Albumin (g/L; mean ± SD)	37.63 ± 5.76 **^+^	33.36 ± 4.47	29.61 ± 4.61
Cholesterol (mg/dL; mean ± SD)	206.88 ± 58.75	173.63 ± 34.42	218.1 ± 69.99
Hb (g/dL; mean ± SD)	11.03 ± 1.27 *^+^	10.27 ± 0.76	10 ± 0.89
CRP (mg/L; mean ± SD)	4.07 ± 3.17 **	5.91 ± 3.36	10 ± 5.80
MAP (mmHg; mean ± SD)	95.11 ± 8.50	94.43 ± 10.23	98.15 ± 13.23
LVH (Yes/No)	4/15 **^++^	15/13 **	12/1

MAP = mean arterial pressure; DM = diabetes mellitus; RD = residual diuresis; Hb = hemoglobin; CRP = C-reactive protein; Kt/V = dialysis adequacy; CrCl = creatinine clearance; D/Pcr = dialysate/plasma creatinine; DPI = dietary protein intake; DEI = dietary energy intake; nPCR = normalized protein catabolic rate; DSR = dialytic sodium removal; USR = urinary sodium removal; TSR = total sodium removal. * *p* < 0.05, ** *p* < 0.01 vs. DCG—High. ^+^
*p* < 0.05, ^++^
*p* < 0.01 vs. DCG—Medium.

**Table 3 metabolites-14-00460-t003:** Comparison of the investigated parameters depending on the DPI lower or greater than 0.8 g/kg/day.

	DPI < 0.8 g/kg/day (n = 18)	DPI > 0.8 g/kg/day (n = 42)	*p*-Value
Age (years; mean ± SD)	57.67 ± 9.25	63.59 ± 11.61	0.184
PD duration (months; mean ± SD)	9.78 ± 10.18	37.11 ± 34.93	**0.011**
BMI (kg/m^2^; mean ± SD)	26.19 ± 2.4	24.02 ± 3.13	**<0.041**
%LBM	49.39 ± 14.73	47.15 ± 11.25	0.689
Protein intake (g/day; mean ± SD)	60.77 ± 6.26	74.3 ± 13.44	**<0.008**
nPCR (g/kg/day; mean ± SD)	0.66 ± 0.1	0.92 ± 0.2	**<0.001**
DEI (kcal/kg/day; mean ± SD)	24.22 ± 4.91	28.89 ± 5.78	**<0.035**
Sodium intake (g/day; mean ± SD)	2.94 ± 0.86	3.69 ± 0.71	**<0.018**
Kt/V	2.11 ± 0.51	2.35 ± 0.47	0.222
CrCl (L/week; mean ± SD)	74.22 ± 30.93	77.36 ± 19.68	0.331
RRF (mL/min/1.73 m^2^; mean ± SD)	4.53 ± 3.38	3.67 ± 2.42	0.557
Albumin (g/L; mean ± SD)	38.44 ± 3.94	37.73 ± 4.9	0.674
Cholesterol (mg/dL; mean ± SD)	212.68 ± 47.18	191.8 ± 58.78	0.367
Peritoneal protein loss (g/day; mean ± SD)	6.3 ± 2.81	8.6 ± 4.99	0.242

BMI = body mass index; LBM = lean body mass; RRF = residual renal function; Kt/V = dialysis adequacy; CrCl = creatinine clearance; DPI = dietary protein intake; DEI = dietary energy intake; nPCR = normalized protein catabolic rate.

**Table 4 metabolites-14-00460-t004:** Univariate associations between protein intake and nutrition parameters.

		DPI (g/kg/day)		Protein Intake (g/day)	
		R	*p*-Value	R	*p*-Value
BMI (kg/m^2^)	Male	−0.434	**0.021**	−0.138	0.484
	Female	−0.365	**0.04**	−0.125	0.496
nPCR (g/kg/day)	Male	0.352	0.067	0.241	0.217
	Female	0.455	**0.009**	0.282	0.117
LBM (%)	Male	−0.204	0.297	−0.053	0.788
	Female	−0.124	0.439	−0.063	0.734
Total energy intake (kcal/day)	Male	0.243	0.213	0.409	**0.031**
	Female	0.255	0.160	0.476	**0.006**
DEI (kcal/kg/day)	Male	0.557	**0.002**	0.355	0.064
	Female	0.655	**0.00**	0.343	0.054
Albumin (g/L)	Male	0.042	0.831	−0.048	0.808
	Female	−0.002	0.991	0.059	0.746

DPI = dietary protein intake; BMI = body mass index; nPCR = normalized protein catabolic rate; LBM = lean body mass; DEI = dietary energy intake.

**Table 5 metabolites-14-00460-t005:** The relationship between LBM and other investigated clinical and laboratory parameters.

	LBM%	
	R	*p*-Value
Female gender	−0.433	**0.015**
Age (years)	−0.261	0.157
BMI (kg/m^2^)	0.462	**0.009**
nPCR (g/kg/day)	0.186	0.316
Davies comorbidity grade	−0.193	0.299
RRF (mL/min/1.73 m^2^)	0.616	**0.000**
Kt/V	0.09	0.636
CrCl (L/week)	0.482	**0.007**
Albumin (g/L)	0.515	**0.003**
Peritoneal protein loss (g/day)	0.133	0.501

BMI = body mass index; RRF = residual renal function; Kt/V = dialysis adequacy; CrCl = creatinine clearance.

**Table 6 metabolites-14-00460-t006:** Correlation analysis between protein metabolism indices and TSR.

	DPI (g/kg/day)		Total Protein Intake (g/day)		nPCR (g/kg/day)	
	R	*p*-Value	R	*p*-Value	R	*p*-Value
TSR (mmol/day)	0.154	0.408	0.267	0.147	0.179	0.336

## Data Availability

The raw data supporting the conclusions of this article will be made available by the authors without undue reservation. Further inquiries can be directed to the corresponding author.

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
