# Peer review of "Estimating Dietary Protein and Sodium Intake with Sodium Removal in Peritoneal Dialysis Patients"

_metabolites, 2024, doi:10.3390/metabo14080460_

Round 1

Reviewer 1 Report

Comments and Suggestions for Authors

This study describes an important dataset which is relevant for the biomedical field. Authors should make certain improvements in the manuscript to get it accepted for publication. Here are the comments:

1. Title should be more scientifically written.

2. Authors should describe the significance of this work clearly. Perhaps can be included in introduction.

3. Some key results must be described in the form of figures (eg association of protein intake vs sodium intake).

4. Methods section should rearranged for easy reading for authors. It is very complicated for readers.

Comments on the Quality of English Language

Overall, English writing in this manuscript should be improved.

Reviewer 2 Report

Comments and Suggestions for Authors

Review Report of manuscript ID Number” metabolites- 3131124”

The research article entitled “Does sodium intake really correlate with protein intake in peritoneal dialysis patients?” submitted for publication in “Metabolites” with manuscript ID Number” metabolites-3131124” has good healthy and comprehensive results and the manuscript is well designed and presented and contains all the necessary experiments to prove his hypothesis.

My general comments on the manuscript are incorporated as follows:

  1. Sample Size: The study's sample size of 60 patients is relatively small. This limitation can affect the generalizability of the findings and may not adequately represent the broader PD patient population.
  2. Cross-Sectional Design: Being a cross-sectional study, it captures a snapshot in time rather than longitudinal data. This design limits the ability to infer causation and the temporal relationships between variables.
  3. Single-Center Study: Conducting the study at a single center limits the diversity of the patient population and may introduce center-specific biases. Multi-center studies are generally preferred for greater external validity.
  4. Measurement Methods: The methods used to measure sodium intake, sodium removal, DPI, nPCR, and LBM should be critically evaluated. If not standardized and validated, these measurements can introduce inaccuracies.
  5. Confounding Factors: The study mentions correlations but does not extensively discuss potential confounders. Factors such as medication use, hydration status, dietary habits, and other comorbid conditions could influence the results and should be controlled for in the analysis.
  6. Lack of Longitudinal Data: Without longitudinal data, the study cannot assess the long-term impact of DPI and sodium intake on cardiovascular morbidity and other clinical outcomes in PD patients.
  7. Nutritional Recommendations: While the study suggests monitoring sodium intake and removal with increased protein intake, it does not provide specific guidelines or consider the practicality and patient adherence to such recommendations.

Scope for Improvement

  1. Increase Sample Size and Diversity: Future studies should include a larger and more diverse patient population to enhance the generalizability of the findings.
  2. Longitudinal Design: A longitudinal study design would help establish causal relationships and assess the long-term impact of dietary interventions on patient outcomes.
  3. Multi-Center Collaboration: Collaborating across multiple centers would reduce center-specific biases and increase the robustness of the findings.
  4. Comprehensive Confounding Control: Thoroughly identifying and controlling for confounding factors would strengthen the validity of the results.
  5. Detailed Nutritional Guidelines: Developing specific and practical dietary guidelines based on the study findings would be beneficial for clinical practice.

Conclusion

The study presents important insights into the relationships between DPI, sodium intake, and patient outcomes in PD patients. However, its small sample size, cross-sectional design, single-center setting, and potential confounding factors limit the strength of its conclusions. Addressing these limitations in future research would provide more robust and actionable data to inform clinical practice

My specific comments on different sections of article are as below:

Abstract:

Some grammatical mistakles in the abstract are followings:

The phrase "Increase of dietary protein intake (DPI) carries a risk for increased sodium intake which further leads to development of cardiovascular morbidity" should have a comma after "intake" and before "which."

The term "USR" should be defined at its first mention, as should "nPCR" and "LBM."

 The sentence "The value of RRF<2ml/min/1.73m2 was associated significantly with lower USR (p=0.000) and lower %LBM (p<0.001)" should use "significantly associated" instead of "associated significantly."

  The phrase "Dialytic and urinary sodium removal (USR) might be indicators of sodium intake" is unclear. It could be rephrased to clearly state that these are potential indicators of sodium intake in patients.

  The sentence "Comparing to patients with DPI<0.8g/kg, patients with DPI>0.8 kg/kg have greater sodium intake (3.69±0.71 vs. 2.94±0.86; p<0.018) and greater nPCR (p=0.023)" should use "Compared to" instead of "Comparing to." Additionally, the units for DPI should be consistent (either "g/kg" or "g/kg/day").

  The use of "significantly" is redundant in multiple instances and should be used only where it adds clarity to the findings.

  The sentence "Protein intake was significantly correlated with sodium intake (p=0.041), but not with total sodium removal (TSR), while strong correlations was noticed between sodium intake and TSR" could be more concise and clearer, such as: "Protein intake was significantly correlated with sodium intake (p=0.041), but not with total sodium removal (TSR). A strong correlation was observed between sodium intake and TSR."

Ensure that all abbreviations and symbols are defined at their first use. For instance, "RRF" should be spelled out as "residual renal function" when first mentioned.

The use of "<" in "RRF<2ml/min/1.73m2" and "DPI<0.8g/kg" should be spaced appropriately for readability, i.e., "RRF < 2 ml/min/1.73m²" and "DPI < 0.8 g/kg."

Technical and Specificity Issues:

The abstract lacks specificity in some areas. For instance, "protein metabolism indices such as nPCR and lean body mass (LBM)" could clarify what these indices measure or indicate.

The mention of "single TSR values were different from corresponding single sodium intake values" is vague. The nature and significance of these differences should be clarified.

Results Table 1

Some variables may require additional context or explanation regarding their clinical significance. For example, while a significant difference in %LBM or RRF is noted, its clinical relevance and implications for patient outcomes should be discussed.

The results for variables such as "Cholesterol," "CRP," "Albumin," "Protein intake," and "Sodium intake" are not statistically significant. While these findings are important, it would be useful to discuss why these results might be clinically relevant despite the lack of statistical significance.

The use of mean ± SD is standard, but for skewed distributions, median and interquartile range (IQR) might be more appropriate.

The inclusion of percentages for categorical variables, such as gender or comorbidity grade, without the corresponding raw numbers (n) can be misleading, especially with small sample sizes.

The use of mean ± SD is standard, but for skewed distributions, median and interquartile range (IQR) might be more appropriate.

The inclusion of percentages for categorical variables, such as gender or comorbidity grade, without the corresponding raw numbers (n) can be misleading, especially with small sample sizes.

The lack of significant differences in some parameters like "BMI," "Protein intake," and "nPCR" suggests either a homogenous population or insufficient sample size to detect differences. This should be addressed, possibly by discussing potential reasons or implications for clinical practice.

The table does not indicate any adjustments made for confounding variables, which could potentially influence the results. Including a note on whether adjustments were made would strengthen the findings.

Table 2:

The headings are not uniformly formatted, leading to inconsistency. For instance, "DCG – Low," "DCG – Medium," and "DCG – High" should be consistently aligned and formatted.

There are inconsistencies in spacing and alignment of values, particularly in columns under clinical parameters and corresponding p-values. This inconsistency affects readability.

The number of decimal places should be consistent throughout the table for clarity. For instance, "CrCl (L/week; mean±SD)" has varying decimal places across groups, which should be standardized.

Some p-values indicate significant differences, but without clear indication of the statistical test used (e.g., ANOVA, t-test), it’s challenging to interpret the results properly.

For parameters with more than two groups and significant p-values (e.g., Kt/V, CrCl), post-hoc analysis results should be reported to identify which groups differ significantly from each other.

Ensure units of measurement are consistently used and clearly stated, especially for parameters like "MAP (mean±SD)" where units (mmHg) are missing.

For significant differences, especially in clinical parameters like "Albumin," "Hb," "CRP," and "LVH," the table should include a brief note or reference to text explaining potential clinical implications or the need for further research.

The analysis should account for potential confounding factors that could influence the results. For example, differences in age or PD duration between groups could impact other clinical parameters.

Table 3

The sample sizes for the groups (N=18 for DPI<0.8g/kg and N=42 for DPI>0.8g/kg) suggest an imbalance. This could impact the power of the statistical tests and should be addressed, either in the text or table.

It is unclear if there are any missing data or if the reported means and SDs are based on all available data. Clarifying this would enhance the credibility of the data.

Table 4

The table reports correlation coefficients (R) and corresponding p-values. However, the choice of correlation type (e.g., Pearson, Spearman) is not specified, which is critical for interpreting the results.

There is a lack of consistency in reporting the statistical significance. Typically, significant p-values are marked (e.g., with an asterisk), but this table lacks such clear markers, making it difficult to identify significant results at a glance.

The units of measurement for each parameter should be clearly stated, particularly for values like "Total energy intake" and "Sodium intake."

The table presents univariate associations without accounting for potential confounding variables. This limits the ability to infer causality or even strong association from the correlations reported.

Table 5

The table includes both positive and negative correlations, such as the negative correlation between female gender and LBM%. The implications of these negative correlations are not discussed or explained, leaving their relevance unclear.

The choice of variables included in the table, such as Davies comorbidity grade and Kt/V, should be justified. It's important to explain why these specific parameters were investigated in relation to LBM%.

Table 6

The reported correlation coefficients are relatively low, indicating weak associations between protein metabolism indices and TSR. The significance and relevance of these weak correlations should be discussed.

Discussion:

The discussion needs to be further updated and revised in terms of describing valuable results and statistical analyses.

Conclusion

The conclusion implies a causal relationship between protein intake and sodium intake/removal but does not provide evidence or mechanisms for this relationship. It is important to clarify whether this is an observed correlation or if there is a known causal mechanism.

The recommendation for "regular measurement of RRF" is broad and does not consider the individual variability among patients. It would be more helpful to provide guidelines on how often these measurements should be taken and what specific indicators to monitor.

The conclusion suggests "checking of patients regarding the adherence to diet" but does not provide practical guidance on how to implement this. For instance, are there specific tools, questionnaires, or methods that should be used to assess dietary adherence?

The conclusion does not acknowledge potential confounding factors that could influence the relationships between protein intake, sodium balance, and nutritional status. Factors like age, comorbid conditions, and baseline nutritional status could all play significant roles.

The overall language of the manuscript requires editing, grammatical corrections and rephrasing. Some sections of the review has extracted from others as it is and therefore 28% similarity index while AI is too low which is a healthy sign about the originality of the manuscript. 

Addressing these comments will definitely improve the level of this article.

Comments on the Quality of English Language

Extensive Engliush editing is required.

Round 2

Reviewer 2 Report

Comments and Suggestions for Authors

The author has carefully addressed all the comments and revisied the manuscriot according to my line of direction. I am satistifed with his efforts in justifying all raised concerns, therefore, I recommend the pubication of this artcile.